# Is the Proof in the Pain? Association between Head and Neck Pain and Vessel Pathology at Follow-Up in Cervical Artery Dissection: A Retrospective Data Analysis

**Jil Baumann [1,†], Miranda Stattmann [1,†] and Susanne Wegener [1,2,*]**

1   Clinical Neuroscience Center, Department of Neurology, University Hospital Zurich,
    8057 Zürich, Switzerland; jil.baumann@uzh.ch (J.B.); miranda.stattmann@usz.ch (M.S.)
2   Medical Faculty, University of Zurich, 8006 Zurich, Switzerland
*   Correspondence: susanne.wegener@usz.ch
†   These authors contributed equally to this work.

**Abstract:** Unilateral head and neck pain is a hallmark of cervical artery dissection (CAD). While pain is conceived as an alarming sign for patients and often leads to discovery of the dissection, it is not known if persistence of pain is associated with the course of CAD. Potentially, pain could indicate persisting vessel pathology and thus guide treatment decisions aimed at reducing risk of ischemic stroke in CAD. We performed a retrospective analysis of data from patients with CAD treated at the University Hospital Zurich (USZ). Only patients with information about the presence of pain, independence after CAD according to the modified Rankin scale (mRS), and imaging-based information on vessel status were included. Patients were grouped according to presence/absence of head and/or neck pain on admission and at a three-month follow-up. We used descriptive statistics and logistic regression to reveal a potential association between pain on admission and pain at follow-up with status of the dissected vessel at follow-up (open vs. stenosed or occluded). We screened 139 patients with CAD between 2014 and 2019 and included 68. Fifty-nine patients (86.8%) had pain on admission, which was resolved in 46 (68%) at follow-up. Our post hoc analysis revealed that more patients with headache or neck pain on admission had a migraine diagnosis in medical history ($n = 7$ (10.4%) vs. $n = 0$ (0%), $p = 0.029$) and that NIHSS on admission was higher in patients with no pain at presentation (group B NIHSS = 3, IQR 8 vs. group A NIHSS = 2, IQR 5, group C NIHSS = 0, IQR 2, $p = 0.041$). There were no other differences between the three patient groups in the descriptive analysis. Logistic regression analysis for vessel status at follow-up did not show an association with pain on admission or at follow-up. In our cohort of patients with CAD, headache was a common initial clinical presentation, which rarely persisted for three months. Headache on admission or at follow-up did not predict persisting vessel pathology in patients with CAD.

**Keywords:** stroke; ischemic stroke; dissection; vessel occlusion; pain; headache

## 1. Introduction

Cerebral artery dissection (CAD) is the leading cause of stroke in younger patients [1,2]. Since the first description in the literature in the 1970s by Miller-Fisher, our understanding of the clinical presentation, etiology, treatment, and outcome of CAD has evolved [1]. Dissection arises from an injury of the outer layer of arteries followed by hematoma within the arterial wall, resulting in extension of the vessel wall or stenosis/occlusion of the inner vessel lumen. Etiology is either spontaneous or secondary to trauma [3]. CAD usually affects the internal carotid artery (ICA = 70%–80%) or the vertebral artery (VA = 15%) [4]. Location of the CAD can be extra- or intracranial; however, it is most commonly in proximity of the artery with a bony structure (e.g., V1 segment of the VA or cervical segment of the ICA) [5].

Considering the risk of developing ischemic stroke in the weeks after CAD, early diagnosis and treatment initiation is of the highest importance [6]. MRI with MRA, CT with CTA, and ultrasound imaging are the diagnostic modalities of choice to diagnose vessel patency [7]. Follow-up imaging is important to recognize thrombus formation, persisting arterial occlusion, clustering of dissections, as well as recurrence, which guides treatment decisions such as the duration of antithrombotic treatments. Based on recent data showing no superiority of antiplatelet drugs or anticoagulants for stroke prevention in CAD, there is no clear consensus on the choice of preventive treatments, which is reflected by European guidelines [8,9].

Headache, along with neck pain, is the leading symptom of CAD: in a recent systematic review, 70% of CAD patients had pain on presentation [1]. In the acute setting, other signs such as ipsilateral Horner's syndrome, tinnitus, vertigo, and nausea can point the clinician towards the correct diagnosis [1]. However, in a large fraction of patients, headache is the only symptom in the initial consultation [10].

So far, the existing literature describes heterogeneous pain characteristics in CAD with considerable differences between patients. A recent study examined almost 300 subjects, confirming head and neck pain as the leading symptom of spontaneous CAD, with a predominance of pulling pain of a novel character [11]. Resolution of pain showed a median of 13.5 days. According to the literature, recanalization of occluded vessels takes longer, usually occurring within six months [12].

So far, no study has examined whether persisting headache associated with CAD correlates with vessel occlusion at follow-up imaging.

## 2. Materials and Methods

### 2.1. Patients

We performed a retrospective analysis of data from the Swiss Stroke Registry of the University Hospital Zurich (USZ) using records dating from January 2014 until December 2019. Patients were screened for cervical artery dissection ($n = 139$) and consented to data usage for research according to the ethics protocol KEK-ZH 2014-0304. After review of the data from the SSR, missing data were completed from the electronic patient records (KISIM, clinical information system USZ). The data were evaluated for record of pain status at initial presentation, follow-up, imaging data at follow-up, death, and correct diagnosis. The follow-up period was defined as the period until 12 months after initial diagnosis of cervical artery dissection. A definite diagnosis of CAD and clearly documented information about the presence of pain both on admission and at follow-up within the 12-month time range was crucial for this retrospective analysis, so all patients for whom this information was not available were excluded. We also excluded patients who declined consent for retrospective analysis of their routine clinical data. Routine follow-up at our clinic usually takes place 3 months after CAD; however, to increase the number of subjects for this analysis, we chose to extend the time window for follow-up to 12 months. A 3-month mRS was chosen as a measure of clinical outcome as this is routinely assessed at the 3-month follow-up appointment after cerebrovascular disease and CAD and is also used as an outcome parameter in other stroke studies. Our protocol was an ad hoc analysis of data based on a headache specialist's viewpoint and not based on previous studies.

### 2.2. Statistical Analysis

We performed the statistical analysis using IBM SPSS Statistics Version 29. We compared groups of patients based on demographic characteristics, clinical vital signs and scores, comorbidities, previous medical history and cardiovascular risk factors, headache, and vessel status.

We present the median values and interquartile range of continuous variables, while nominal variables are shown as percentages. Patients were split into groups according to headache at initial presentation versus headache at follow-up (Group A = Pain +/−, Group B = Pain −/− and Group C = Pain +/+). Considering the small sample size, we

compared groups using Chi-squared and Kruskal–Wallis tests for nominal and continuous variables, respectively. Consequently, the Chi-squared test was applied to sex, location of dissection, vessel pathology and persisting vessel pathology, medical treatment, outcome, and all points listed for medical history. The Kruskal–Wallis test was used for linear data (age, BMI, blood pressure, days to follow-up) or pseudo-linear data (NIHSS, mRS). We tested normal distribution of linear data. There was normal distribution for some of the linear data in some of the subgroups (age and BMI for all groups, NIHSS for group B, blood pressure for groups B and C, time between onset and follow-up for group B), but not in all (see Supplemental Table S1). We did not find normal distribution for NIHSS in groups A and C, blood pressure in group A, time between onset of pain and follow-up in groups A and C, and 3-month mRS in all groups. Due to this and the different group sizes, we decided to use more conservative non-parametric tests, such as the Kruskal–Wallis test, that do not assume normal distribution of data. Additional statistical output for the Kruskal–Wallis statistics can be found in Supplemental Table S2.

In addition, we performed a binary logistic regression analysis for vessel status at follow-up (either normal or stenosed/occluded) with the covariates of pain, initial vessel status, age, sex, and hypertension. $p$ values (representing exact 2-sided significance) were considered statistically significant if $p$ was <0.05.

## 3. Results

We included 68 patients after screening for missing data and applying exclusion criteria (Figure A1). In our patient collective, median age was 48.6 (IQR 18) and 33.8% of the patients were female ($n$ = 23, Table A1).

In total, 86.8% of our patients reported headache initially ($n$ = 59). Of these, 68% presented with headache at onset and reported resolution at follow-up ($n$ = 46, Group A), 13% never reported having any headache ($n$ = 9, Group B), and 19% had persistent headache ($n$ = 13, Group C). We found evidence of hyperlipidemia in 50% ($n$ = 34) and hypertension in 27.9% ($n$ = 19) of our patients. Smoking was reported by 20.6% ($n$ = 14). Overall, comorbidities were rare.

We found no significant difference between the groups regarding comorbid coronary heart disease, prosthetic heart valves, atrial fibrillation, smoking, hyperlipidemia, diabetes, hypertension, and previous stroke (Table A1).

We found that 10.4% ($n$ = 7) of patients had a history of migraine (four with and three without aura, $p$ = 0.029).

In patients with headache, the vessel pathology was usually located on the same side as the headache (68.9%, $n$ = 31), with a majority located in the anterior circulation (57.4%, $n$ = 39). There was no statistically significant difference between the groups regarding the location of the dissected vessel (anterior vs. posterior circulation, $p$ = 0.131), the extent of the vessel pathology (occlusion vs. stenosis, $p$ = 0.303), and presence of multiple dissections ($p$ = 1.000). We found no significant differences in initial systolic blood pressure ($p$ = 0.177); however, severity of clinical presentation measured with the NIHSS was higher in the group without headaches ($p$ = 0.041).

Acute treatment in the majority of patients consisted of antiplatelets (57.4%, $n$ = 39) or anticoagulants (16.2%, $n$ = 11). There was no difference between groups in this regard (63% vs. 44.4% vs. 69.2% $p$ = 0.528 and 17.4% vs. 0% vs. 23.1% $p$ = 0.410, respectively). Treatment with intravenous lysis on admission was initiated in 25% of our patients ($n$ = 17, $p$ = 0.154) and 16.2% received intraarterial treatment ($n$ = 11, $p$ = 0.090). Only one patient from group B suffered recurrent stroke. Outcome as measured by mRS at three months was consistently very favorable, with a median of zero (IQR 1) and no statistically significant difference across all groups $p$ = 0.137, Table A1). At follow-up, we found that over 50% of patients showed resolution of vessel pathology on imaging (52.9%), with no significant difference between groups ($p$ = 0.272). However, we found a statistically significant difference in time to follow-up between groups ($p$ = 0.010).

In binary logistic regression analysis, we found no association between vessel status at follow-up and initial vessel status, age, sex, or hypertension (Table A2).

## 4. Discussion

Pain on admission was common in patients presenting with CAD (86.8%). Similar to other studies, vessel pathology and pain were mostly ipsilateral. Pain persisted until follow-up at three months in only 22% of patients with pain on admission. In our patient collective, we found that persisting pain at follow-up was not associated with persisting vessel pathology. In line with previous studies, approximately 50% of patients showed resolution of vessel pathology at follow-up [13], while in those patients who presented with headache initially (group A and C, $n = 59$), pain resolved in 78% (group A, $n = 46$).

We found no significant differences between the groups regarding the location of the dissected vessel or extent of vessel pathology. Our descriptive analysis showed higher admission scores in those patients presenting without pain ($p = 0.041$). One might speculate that patients who had pain on admission presented to the hospital earlier, leading to earlier and more efficient stroke treatment. However, we found no significant difference between the groups regarding acute stroke treatment. Logistic regression analysis showed no association between vessel status at follow-up and initial vessel status, pain on admission and/or at follow-up, medical history of hypertension, age, or sex. We found a statistically significant difference in time to follow-up between groups ($p = 0.010$), with patients who had no pain at any point in time (group B) presenting the latest for follow-up. Patients with pain at follow-up had the shortest mean time to follow-up, leading us to speculate that persisting pain might make patients seek earlier appointment times.

One reason to suspect pain in relation to CAD would be irritation of perivascular pain-sensitive nerves caused by mechanical compression or direct stimulation of pain receptors in the vessel wall. Supporting this hypothetical mechanism, dissection of the carotid artery seems to cause referred pain to areas also reported as symptomatic during balloon inflation of the carotid artery [14]. Depending on which artery is affected (carotid versus vertebral), as well as the exact anatomical location (extracranial involving the carotid sinus versus intracranial), different afferent nerves transmit pain. While the phenomenon of acute pain may be causally related to disruption of innervation of the vessel wall at the onset of CAD [15], our data do not indicate that once the vessel is dissected, vessel status per se (occlusion, stenosis, or complete recovery) is related to spontaneous pain relief. This is consistent with the fact that intramural hematoma resolution and recanalization of the affected vessel usually take place within three to six months after CAD. Interestingly, studies have found an increase in the external diameter of the artery irrespective of stenosis, which could also irritate surrounding nerves despite vessel patency [16]. Another factor that might contribute to pain in CAD is the presence of collaterals in cerebral circulation, as collateral openings have been shown to be predictors of headache occurrence in patients with stroke [17]. Further studies should take this into account.

Since CAD has previously been associated with a history of migraine, in particular migraine without aura [18], we investigated the relationship between previous migraine diagnosis and CAD in our cohort. Compared to other reports with numbers as high as 30% [18], we found a smaller percentage (10%) of patients with a diagnosis of migraine. Our findings suggest that a previous migraine diagnosis was associated with headache/neck pain at presentation and at follow-up. We found no patients with a previous diagnosis of migraine in the group that had no pain at presentation or follow-up. Altered pain processing pathways during a migraine could contribute to headache development and headache persistence in CAD [19]. However, larger studies with a higher number of patients are needed to confirm this result. There were no statistically significant differences concerning other comorbidities.

Long-term follow-up data on headache after CAD are scarce. However, supporting the findings in our cohort, a recent trial reported persisting headache at follow-up (median time 6.5 year) in 25.6% of patients [11]. Interestingly, this was novel pain, not persistence

of the original headache. In our data set, we had little information on persistence versus recurrence of pain. Qualitatively, two patients reported "intermittent" headache and one spoke of "recurring irregular" pain. Furthermore, it has to be taken into account that our follow-up period was comparatively short. Additionally, records on headache characteristics on admission were missing in many patients. Although it was routinely assessed if pain was present, only around 20% of the affected patients had a detailed record of location and quality of their pain on admission. We feel that history of pain persistence or recurrence should be obtained at follow-up. Pain quality, acuteness of onset, and novel character of headache are crucial information that should alert the clinician to consider further diagnostic testing. This is especially important, as some patients with CAD only report a headache and have no other focal neurologic signs.

In addition to missing data, another limiting factor of our study is the small sample size. We found a median age of 48.6 years and a predominance of men in our group, similar to other big trials. This leads us to believe that our study cohort is representative of the population at large. The representation of patients with dissections of the posterior circulation was rather high at 43% but was in line with other studies [11]. However, our retrospective study design and lack of external validity have to be taken into account. A prospective study with a larger cohort of patients with standardized assessment of headache characteristics both on admission and at follow-up would provide better insight and external validity in further studies. For this explorative analysis, no a priori power calculation was performed, which is another limitation of our study. A prospective study with a larger cohort of patients with standardized assessment of headache characteristics both on admission and follow-up would provide better insight in further studies.

## 5. Conclusions

In conclusion, we found that vessel pathology after CAD at follow-up was independent of headache status. Thus, our findings provide a point of reassurance for patients who might be concerned about the persistence of pain after CAD. Judging from our data, pain does not suggest persisting vessel pathology. However, more studies with a larger cohort of patients are needed to understand the specific type of pain in CAD and the time course of its evolution.

**Supplementary Materials:** The following supporting information can be downloaded at: https://www.mdpi.com/article/10.3390/ctn7020015/s1, Table S1: Test of normal distribution for linear variables; Table S2: Statistics for Kruskal–Wallis testing.

**Author Contributions:** Conceptualization, S.W.; methodology, S.W.; validation, formal analysis, S.W., M.S. and J.B.; investigation, S.W. and J.B.; resources, S.W.; data curation, J.B.; writing—original draft preparation, J.B. and M.S.; writing—review and editing, S.W., M.S. and J.B.; visualization, S.W. and M.S.; supervision, S.W.; project administration, S.W.; funding acquisition, S.W. All authors have read and agreed to the published version of the manuscript.

**Funding:** This research received no external funding.

**Institutional Review Board Statement:** The study was conducted in accordance with the Declaration of Helsinki and approved by the Ethics Committee of Zurich (KEK-ZH 2014-0304).

**Informed Consent Statement:** Only retrospective data collection was performed, and patients were screened for general consent to data usage for research according to the ethics protocol.

**Data Availability Statement:** The data presented in this study are available upon request from the corresponding author. The data are not publicly available due to privacy protection.

**Conflicts of Interest:** The authors declare no conflict of interest.

## Appendix A

**Table A1.** Group characteristics.

| | All $n$ = 68 (100%) | Group A (+/−) $n$ = 46 (68%) | Group B (−/−) $n$ = 9 (13%) | Group C (+/+) $n$ = 13 (19%) | $p$ Value |
|---|---|---|---|---|---|
| Demographic data | | | | | |
| Age, median (IQR) | 48.6 (17) | 48.6 (18) | 55.4 (11) | 47.0 (13) | 0.104 |
| Female sex, $n$ (%) | 23 (33.8%) | 17 (37%) | 1 (11.1%) | 5 (38.5%) | 0.367 |
| Location of dissection, $n$ (%) | | | | | 0.131 |
| Anterior circulation (carotid artery) | 39 (57.4%) | 24 (52.2%) | 8 (88.9%) | 7 (53.8%) | |
| Posterior circulation (vertebral or basilar artery) | 29 (42.6%) | 22 (48%) | 1 (11.1%) | 6 (46.2%) | |
| Initial vessel pathology, $n$ (%) | | | | | 0.303 |
| No vessel pathology | 1 (1.5%) | 0 (0%) | 0 (0%) | 1 (7.7%) | |
| Occlusion | 35 (51.5%) | 23 (50%) | 4 (44.4%) | 8 (61.5%) | |
| Stenosis | 32 (47.1%) | 23 (50%) | 5 (55.6%) | 4 (30.8%) | |
| More than one dissected vessel | 6 (8.8%) | 4 (8.7%) | 1 (11.1%) | 1 (7.7%) | 1.000 |
| Patient characteristic, median (IQR) | | | | | |
| First systolic blood pressure (mmHG) | 141 (25) | 141.0 (29) | 146.0 (33) | 134 (22) | 0.177 |
| BMI, median (IQR) | 23.9 (5) | 24.0 (6) | 23.1 (7) | 23.0 (5) | 0.736 |
| Clinical Scores, median (IQR) | | | | | |
| NIHSS on admission | 1 (5) | 2 (5) | 3 (8) | 0 (2) | 0.041 |
| mRS after 3 months ($n$ = 67) | 0 (1) | 1 (1) | 0 (2) | 0 (1) | 0.137 |
| Treatment, $n$ (%) | | | | | |
| Antiplatelet drugs | 39 (57.4%) | 26 (63%) | 4 (44.4%) | 9 (69.2%) | 0.528 |
| Anticoagulants | 11 (16.2%) | 8 (17.4%) | 0 (0%) | 3 (23.1%) | 0.410 |
| Intraarterial treatment | 11 (16.2%) | 8 (17.4%) | 3 (33.3%) | 0 (0%) | 0.090 |
| Intravenous rTPA | 17 (25%) | 12 (26.0%) | 4 (44.4%) | 1 (7.7%) | 0.154 |
| Outcome, $n$ (%) | | | | | |
| Recurrent stroke | 1 (1.5%) | 0 (0%) | 1 (11.1%) | 0 (0%) | 0.132 |
| Persisting vessel pathology, $n$ (%) | | | | | 0.272 |
| No persisting vessel pathology | 36 (52.9%) | 26 (56.5%) | 3 (33.3%) | 7 (53.8%) | |
| Persisting stenosis | 17 (25.0%) | 10 (21.7%) | 2 (22.2%) | 5 (38.5%) | |
| Persisting occlusion | 15 (22.1%) | 10 (21.7%) | 4 (44.4%) | 1 (7.7%) | |
| Time between onset and pain follow-up (days), median (IQR) | 113 (54) | 116.0 (63) | 140.0 (76) | 94.0 (29) | 0.010 |
| Medical History, n (%) | | | | | |
| Migraine ($n$ = 67) | 7 (10.4%) | 3 (6.7%) | 0 (0%) | 4 (30.8%) | 0.029 |
| Peripheral artery disease | 0 (0%) | | | | |
| Low ejection fraction ($n$ = 62) | 0 (0%) | | | | |
| Prosthetic heart valves | 1 (1.5%) | 1 (2.2%) | 0 | 0 | 1.000 |
| Coronary heart disease | 2 (2.9%) | 2 (4.3%) | 0 | 0 | 1.000 |
| Atrial Fibrillation | 1 (1.5%) | 1 (2.2%) | 0 | 0 | 1.000 |
| Smoking | 14 (20.6%) | 9 (19.6%) | 2 (22.2%) | 3 (23.1%) | 1.000 |
| Hyperlipidemia | 34 (50%) | 25 (54.3%) | 3 (33.3%) | 6 (46.2%) | 0.506 |
| Diabetes | 3 (4.4%) | 1 (2.2%) | 1 (11.1%) | 1 (7.7%) | 0.243 |
| Hypertension | 19 (27.9%) | 15 (32.6%) | 2 (22.2%) | 2 (15.4%) | 0.438 |
| TIA | 0 (0%) | | | | |
| Intracerebral hemorrhage | 0 (0%) | | | | |
| Stroke | 2 (2.9%) | 1 (2.2%) | 1 (11.1%) | 0 (0%) | 0.283 |

Legend Table A1: Patient characteristics (demographic data, clinical scores, and vital parameters), vessel pathology, headache characteristics, and outcome. Data are expressed as number of patients ($n$) and percentages or median and interquartile range (IQR). $p$ values were obtained according to Pearson's Chi-squared test and Kruskal–Wallis test where appropriate. Group A (+/−) = pain on initial presentation, no pain at follow-up; Group B (−/−) = no pain; Group C (+/+) = pain on initial presentation, pain at follow-up.

**Table A2.** Binary logistic regression for vessel status at follow-up.

| KERRYPNX | Regression Coefficient | Standard Error | Wald | df |
|---|---|---|---|---|
| Age | 0.049 | 0.025 | 3.760 | 1 |
| Groups (A = 1/0, B = 0/0, C = 1/1) | 0.276 | 0.335 | 0.679 | 1 |
| Sex | −0.216 | 0.576 | 0.141 | 1 |

**Table A2.** *Cont.*

| KERRYPNX | Regression Coefficient | Standard Error | Wald | df |
|---|---|---|---|---|
| Initial Vessel Status | 0.708 | 0.525 | 1.820 | 1 |
| Medical History of Hypertension | 0.050 | 0.625 | 0.006 | 1 |
| Constant | −3.604 | 1.628 | 4.902 | 1 |

| | Sig. | Exp(B) | 95% confidence interval for EXP(B) | |
| | | | Lower Bound | Upper Bound |
|---|---|---|---|---|
| Age (calc.) | 0.052 | 1.050 | 0.999 | 1.103 |
| Groups (A = 1/0, B = 0/0, C = 1/1) | 0.410 | 1.318 | 0.684 | 2.540 |
| Sex | 0.707 | 0.806 | 0.260 | 2.491 |
| Initial Vessel Status | 0.177 | 2.029 | 0.726 | 5.674 |
| Medical History of Hypertension | 0.936 | 1.051 | 0.309 | 3.581 |
| Constant | 0.027 | 0.027 | | |

Legend Table A2: Binary logistic regression for vessel status at follow-up.

*Patient Flow Chart*

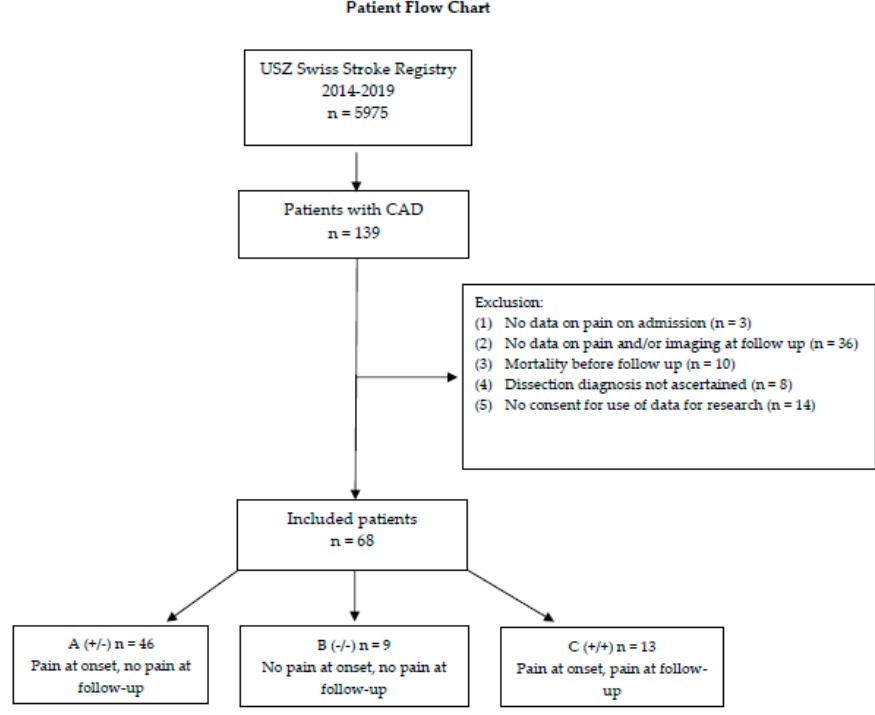

**Figure A1.** We screened 5975 patients with cervical artery dissection and included 68 patients in our analysis. CAD, cervical artery dissection.

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
