# Peer review of "Is the Proof in the Pain? Association between Head and Neck Pain and Vessel Pathology at Follow-Up in Cervical Artery Dissection: A Retrospective Data Analysis"

_ctn, doi:10.3390/ctn7020015_

Round 1

Reviewer 1 Report

It is an interesting retrospective analysis of data from patients with CAD regarding the presence or absence of pain on admission and 3-months follow up. However, it is not clarified whether pain occurred as local pain (localized in the carotid region, the face or the neck) or, on the other hand, as  headache mimicking migraine.  A major limitation of the study is that data are missing about the number of patients with a history of migraine.

Case control studies have revealed that migraine is an independent risk factor of CAD. It should be discussed in more detail how migraine disease might influence the presence of pain at CAD onset and 3 months follow up.

The authors did not show an association between extracranial vessel status and presence of headache on admission or follow up. However, it cannot be ruled out that headache onset especially bilateral in CAD is influenced by the presence of cerebral collateral pathways rather than by the extracranial vessel pathology. 

(!) Please check the references for example in the reference (15) the year of publication is missing  and in (16) the journal is missing.  

Reviewer 2 Report

Moderate changes in grammar are required.

Round 2

Reviewer 2 Report

Minor editing required.
